# Exploring the Impact of 3D Printing Parameters on the THz Optical Characteristics of COC Material

**DOI:** 10.3390/ma17205104

**Published:** 2024-10-19

**Authors:** Mateusz Kaluza, Michal Walczakowski, Agnieszka Siemion

**Affiliations:** 1Faculty of Physics, Warsaw University of Technology, Koszykowa 75, 00662 Warsaw, Poland; agnieszka.siemion@pw.edu.pl; 2Institute of Optoelectronics, Military University of Technology, gen. S. Kaliskiego 2, 00908 Warsaw, Poland; michal.walczakowski@wat.edu.pl

**Keywords:** THz radiation, THz time-domain spectroscopy, cyclic olefin copolymer, 3D printing, fused deposition modeling

## Abstract

In terahertz (THz) optical systems, polymer-based manufacturing processes are employed to ensure product quality and the material performance necessary for proper system maintenance. Therefore, the precise manufacturing of system components, such as optical elements, is crucial for the optimal functioning of the systems. In this study, the authors investigated the impact of various 3D printing parameters using fused deposition modeling (FDM) on the optical properties of manufactured structures within the THz radiation range. The measurements were conducted on 3D printed samples using highly transparent and biocompatible cyclic olefin copolymer (COC), which may find applications in THz passive optics for “in vivo” measurements. The results of this study indicate that certain printing parameters significantly affect the optical behavior of the fabricated structures. The improperly configured printing parameters result in the worsening of THz optical properties. This is proved through a significant change in the refractive index value and undesirable increase in the absorption coefficient value. Furthermore, such misconfigurations may lead to the occurrence of defects within the printed structures. Finally, the recommended printing parameters, which improve the optical performance of the manufactured structures are presented.

## 1. Introduction

The dynamic advancement of terahertz (THz) technology has been recently used across various fields of science and industry. In recent years, there has been an increasing number of scientific studies exploring the potential applications of THz systems in medicine [1,2], telecommunications [3,4], and security systems [5,6]. Such systems require proper steering of THz beams, typically achieved through the use of passive optical elements [7,8]. In many stages of their development, polymer-based manufacturing processes are utilized, ensuring product quality and guaranteeing high maintenance of optical systems. Polymers are increasingly applied in optical systems due to their versatility in shaping and manufacturing, allowing for the creation of lightweight and durable components such as lenses, waveguides, or all other optical elements. Additionally, polymer-based materials can be engineered to have specific optical properties, making them ideal for use in coatings, filters, and different optical elements while maintaining cost-effectiveness and adaptability in design. Due to the fact that THz sources are relatively low-efficient, optical systems often operate at low-power/intensity levels of the THz signals. Therefore, the high optical efficiency of passive optics becomes a crucial factor in minimizing additional unwanted losses within the system. Improving the optical efficiency of these elements can be divided into two main aspects. Firstly, it involves selecting the appropriate type of optical structure to fulfill a specific functionality [9]. Secondly, it concerns the selection of the appropriate type of material that possesses the desired optical properties as well as the choice of the production method for the structure [10,11].

In the THz radiation range, the most commonly used material for passive optical elements production is silicon. It exhibits an exceedingly low absorption coefficient (high transparency) and extremely low dispersion. However, silicon has a relatively high refractive index, approximately 3.4 in the THz radiation range. Thus, it introduces relatively high Fresnel losses to the system (almost half of the incident THz radiation is reflected from the silicon surface). Therefore, the use of such material requires the application of anti-reflection coatings to minimize parasitic reflections and assure efficient functioning [10,12].

This study focuses on the latter aspect. The influence of the 3D printing parameters using fused deposition modeling (FDM) technology on the optical properties of cyclic olefin copolymer (COC) material was investigated. COC is the transparent and biocompatible material. Because of its exceptional THz optical properties, it is considered one of the leading materials for the manufacturing and/or prototyping of THz passive optical components [13]. In optimizing these parameters, the characteristics of the optical properties of COC 3D prints can be effectively improved. So, it can lead to enhancing the optical efficiency of optical components, especially diffractive optical elements (DOEs) manufactured from this material.

## 2. Methodological Requirements

In numerous THz optical systems, the shaping of wavefronts is achieved through the utilization of refractive optics. However, it is important to note that refractive lenses designed for THz radiation have significant thicknesses (especially in cases when focal length starts to be comparable to the diameter of the element) due to the relatively long wavelengths in this radiation range. Additionally, refractive optical elements perform simple functionalities such as altering the form of the wavefront or focusing the beam onto a detector. Due to the high complexity of various optical systems, diffractive optical elements are increasingly being utilized, allowing for the arbitrary manipulation of optical beams depending on the design structure [8,14]. DOEs ensure significantly higher optical efficiency for designed frequencies and have significantly reduced thickness compared to refractive optics (the thickness of a DOE is in the order of the radiation wavelength).

To achieve complex functionalities in optical systems, DOEs may be designed using iterative algorithms [15] (e.g., based on the Gerchberg–Saxton algorithm) or neural networks [16,17]. Consequently, generated phase distributions representing phase delay profiles of DOEs are often complex. Fortunately, innovative spatial modeling methods enable the precise spatial representation of complex phase maps [18], and additive manufacturing technologies serve as optimal tools for prototyping DOEs that implement various functionalities in optical systems. In 2016, Furlan et al. presented various types of focusing diffractive lenses for a frequency of 0.625 THz, manufactured using selective laser sintering (SLS) technology from polyamide 6 (PA6) material [19]. In the same year, Liu et al. demonstrated DOEs for an accelerating THz Airy beam propagating in free space, fabricated using PolyJet technology with a photopolymer material known commercially as VeroWhitePlus [14]. In 2019, Luo et al. presented a study on a diffractive component for processing a continuum of wavelengths from a broadband source using a neural network approach. The DOE was manufactured using a material commercially known as VeroBlackPlus RGD875, indicating the use of PolyJet technology [17]. In 2023, our study presented a DOE for the spatial demultiplexing of THz signals in the frequency domain, where the structures were fabricated from highly transparent COC material using FDM technology [20]. This arises from the fact that the additive manufacturing methods provide the relatively high spatial resolution (compared to THz wavelengths), which reproduces the spatial models of complex phase profiles accurately. Therefore, the manufactured DOEs exhibit relatively high quality and functionality that strictly align with the numerical simulations [20].

It is also worth mentioning that recent studies have increasingly focused on the additive manufacturing of DOEs and the optical properties of the polymeric materials used in these technologies [21,22]. In recent years, studies have shown that some polymeric materials possess suitable optical properties for the fabrication of DOEs, comparable to polytetrafluoroethylene (PTFE), which is predominantly used in commercially available lenses. The materials that exhibit a low absorption coefficient in the THz radiation range are identified as polypropylene (PP) [11,23,24,25], high-impact polystyrene (HIPS) [11,23,25], high-density polyethylene (HDPE) [23,25], styrene butadiene copolymer (SBC)(e.g. BendLay) [24,25], and COC (e.g. TOPAS) [26,27]. In all the mentioned studies, the investigated samples were fabricated using additive manufacturing with FDM technology. However, there are no studies in the literature on detailed investigations of the impact of 3D printing parameters on the optical properties of printed objects. It is also worth mentioning that polymer-based composites allow for the modification of the optical properties of the polymer base material, ensuring that they can be tailored to meet specific application requirements [24]. Additionally, Busch et al., in their study, demonstrated the differences in the optical properties of COC material samples fabricated using injection molding and 3D printing with FDM technology [26]. The highly transparent and biocompatible COC is a very promising candidate when considering “in vivo” applications in case when contact with living tissues might be necessary [28]. The studies on the optical properties of materials in the THz range are conducted using THz time-domain spectroscopy (THz TDS) [24,29]. This technique measures both the attenuation and phase delay introduced by the material (sample), enabling the determination of its absorption coefficient and refractive index. In this method, THz pulses are generated and detected using photoconductive antennas, and the time-domain analysis of the signal provides amplitude and phase information across a wide frequency range, typically from 100 GHz to 4 THz [30,31]. Considering THz diffractive optics, the desired optical properties of materials are a low absorption coefficient and refractive index enabling us to design relatively thin elements without large Fresnel losses. Therefore, the radiation attenuation introduced by the material should be minimized by the choice of proper material. Additionally, such material should have the refractive index ranging from 1.45 to 1.65, which seems to be optimal as significant Fresnel reflection losses do not occur. For such refractive index values, the illuminated designed structures introduce the appropriate phase delay within reasonable thickness. Moreover, the thickness of the 3D model can be represented as the several dozen layers of the 3D printed element in the manufacturing process. Thus, the manufactured structure possesses a quasi-continuous phase profile even when it is designed for such a high frequency of 1 THz [32].

Selecting appropriate printing parameters for DOE production using additive manufacturing methods is crucial. Primarily, each manufactured DOE should be homogeneous throughout its entire volume. Polymer filament manufacturers in FDM technology focus on presenting material specifications with the proposed printing parameters tailored for non-technological applications. This implies that the material should not deform or leave artifacts during printing. The standard infill density of the print recommended by manufacturers ranges from 10% to 40%. However, to obtain a high homogeneity of the DOE structure, the infill density value of the print needs to be equal to 100%. Additionally, particular attention should be paid to the precision of the connections of the adjacent lines and layers during printing. This is because the imprecise printing lines and layer connections may lead to unwanted diffraction phenomena at higher frequencies (especially when the lines cease to have sub-wavelength- and wavelength-size dimensions). It is essential to emphasize that setting the appropriate printing parameters is non-trivial, as the viscoelastic polymers exhibit non-Newtonian fluid behavior and are subject to the Barus effect and normal stress effect during flow through the nozzle used in FDM 3D printing technology [33,34]. Thus, research on the printing parameters aims to determine the desired parameters enabling the correct fabrication of optical elements such as DOEs and improving their optical properties compared to structures produced with the parameters proposed by the material manufacturer. Polymer-based manufacturing processes play a crucial role in the development of THz optical systems, enabling the creation of lightweight, durable, and versatile components such as lenses, filters, polarizers, waveguides, and many more products, while ensuring product quality, specific optical properties, and cost-effectiveness.

The novelty of this study lies in conducting a detailed research on the impact of various 3D printing parameters on the optical properties of DOE structures printed from COC material in the THz radiation range. Additionally, specific printing parameters are proposed to ensure high efficiency by minimizing losses caused by incorrect parameter selection. It is important to emphasize that COC material is highly transparent in the THz radiation range and is increasingly used in the production of THz diffractive optics, as evidenced by numerous scientific studies demonstrating structures made from this material.

## 3. Materials

To investigate the influence of the various printing parameters on the THz optical properties of COC material, multiple series of samples for THz TDS evaluation were produced using FDM technology with varying printing settings. Each series of the samples was produced by changing one of the parameters of 3D printing and later comparing them to the reference sample. The printing parameters that were verified include the temperature of the nozzle, the size of the nozzle, the printing speed, the width of the extruded lines, the fan speed (the level of cooling used in the printing process), the material flow ratio, the height of the print layers, and the infill pattern shape.

In this study, objects were manufactured using a Prusa MK3S+ printer (Prusa Research, Prague, Czech Republic), primarily utilizing a hardened steel nozzle with electroless nickel plating and a tungsten disulphide fullerene structure (WS_2_) nanoparticle coating, featuring a diameter of 400 μm. An exception was made for the set of measurements evaluating the influence of the nozzle diameter, for which a brass nozzle with a 250 μm diameter and a hardened steel nozzle with electroless nickel plating and WS_2_ nanoparticle coating with a diameter of 600 μm were used. The horizontal accuracy of the printer is approximately 10 μm, while the vertical accuracy is approximately 2 μm. The filament used for 3D printing was COC with a diameter of 1.75 mm and a transparent color, manufactured by Creamelt in Switzerland (batch number 210416).

For the experimental evaluation of the 3D printed samples, the TPS 3000 spectrometer (TeraView, Cambridge, UK) was utilized. The spectrometer provides a spectral measurement range of 0.1 THz to 4 THz, with a signal-to-noise ratio exceeding 70 dB at the peak and a spectral resolution of approximately 6 GHz. It operates in rapid-scan mode, performing 30 scans per second with 3000 averages per measurement, resulting in a total measurement time of approximately 2 minutes per sample. The system was purged with dry air to reduce the relative humidity to below 1%. The upper limit of the spectral range is influenced by the properties of the sample, particularly its attenuation.

Two samples were produced for each setting of each investigated printing parameter. However, reference samples were manufactured and measured separately within each experimental group. The printing parameters of the reference samples were preliminarily tested in detail, resulting in the proper printing of the DOE structures that exhibited the appropriate performance in the optical setups. The DOEs manufactured using the 3D printing parameters corresponding to the reference structures were presented in our previous studies [16,20]. This study aimed not only to verify the previously determined printing parameters considered as reasonably correct but also to assess their influence on the optical properties of the produced DOE structures when deviating significantly from them. The reference sample was manufactured with a nozzle diameter size equal to 400 μm, a nozzle temperature of 240 °C, a printing speed of 60 mm/s (but 20 mm/s for the first layer of the print), an extruded line width of 450 μm, no cooling (0% cooling level), a material flow ratio of 102% of the standard value, a layer height of 100 μm, and a rectilinear infill pattern. All the samples (except the sample manufactured in the horizontal position) were designed as the cylindrical objects with the base diameter equal to 35 mm and the height of 4 mm. This method of sample design allows for the accurate characterization of its optical properties in the central part of the sample, such as through achieving the appropriate printing speed when the nozzle accelerates from the edge of the sample. Additionally, the carefully selected thickness of each sample enables shifting, in time, the parasitic Fresnel reflections inside the samples. For smaller sample thicknesses, this additional peak does not allow us to properly determine the complex permittivity of the material. The 3D model of the cylindrical sample is presented in Figure 1a. Figure 1b–d illustrate the sliced models of samples designed with various infill patterns. In this study, all samples were sliced using PrusaSlicer version 2.3.1. The colors of the sliced objects indicate the predicted printing speeds achieved by the nozzle during the printing process.

The sliced models of the cylindrical sample shown in Figure 1 demonstrate that only the (aligned) rectilinear infill pattern allows for achieving the targeted constant printing speed of 60 mm/s in the middle part of the sample. Therefore, all samples in this study (except for the investigation of the influence of infill patterns on the optical properties of manufactured objects) were designed with rectilinear infill patterns. All the samples, except those used for investigating line thickness parameters, were printed with a first-layer thickness of 200 μm, followed by 38 layers at 100 μm each. The final thickness of the samples also depends on the properties of the COC material, such as internal stress. However, when the printing parameters were correctly configured, the total thickness of the samples varied by no more than 50 μm from the target thickness of 4 mm, with no significant defects observed, as noted in the reference samples. Photographs of the reference samples and the selected samples with deformations (marked in red), obtained for certain printing parameters, are presented in Figure 2.

The photographs of the reference samples manufactured using FDM 3D printing technology are presented in Figure 2a (top view) and Figure 2b (view from an angle). The reference samples were manufactured with the preliminary tested parameters. The significant deformations of the samples are not visible. Figure 2 also illustrates how the incorrect printing parameters can affect the shape and introduce deformations to the 3D printed objects. Figure 2c (top view) and Figure 2d (view from an angle) show photographs of the samples manufactured with the exceedingly thick layers. The applied layer thickness in the printing process was equal to 400 μm. In the manufacturing process of such thick layers, the printing speed was reduced to 50% of the reference value. The inappropriate layer thickness had a significant impact on the accuracy of the lines’ and layers’ bonding of the printed object. It led to a reduction in the homogeneity of the sample’s structure, resulting in deformations (marked in red) that are clearly visible in the central part of the samples. Figure 2e (top view) and Figure 2f (view from an angle) depict the photographs of the samples printed with overly high material flow rate, which was equal to 110% of the standard value. The overflowing material resulted in the deformations (marked in red) that are visible on the top surface and the side walls of the samples. Except the deformations on the surface of the structure, the inaccurate connection of the lines and the wavy layer bindings can be observed.

Subsequently, samples with various types of infill patterns were printed and subjected to optical property analysis using THz TDS. Images of the samples are presented in Figure 3.

Figure 3 illustrates the samples manufactured with various infill patterns. The analyzed patterns examined in this study include the rectilinear, aligned rectilinear, concentric, octagram spiral, and the cuboid sample printed in the vertical position with a rectilinear infill pattern, as presented in Figure 3, with the notation (a) to (e), respectively. The initial visual comparison indicates the differences in the homogeneity of the samples, e.g., the sample printed with an aligned rectilinear infill pattern exhibits the most transparent properties for visible light. However, it needs to be emphasized that illustrated photographs were obtained with a camera dedicated to visible light image registration. The THz radiation exhibits significantly longer wavelengths, ranging in millimeters. Thus, the visible deviations from homogeneity might not affect the THz optical performance of the samples. For the longer wavelengths, the sub-wavelength imperfections and patterns of the structures have no direct impact. However, the deformation of the samples or the incomplete infill of the samples will be clearly visible in optical property verification using the THz TDS method.

The THz optical properties of the 3D printed samples using biocompatible, highly transparent COC material were experimentally verified using THz TDS. The samples were inserted in a chamber with pumped dry air in the broadband-focused THz beam. In this way, the influence of the water vapor absorption, which is present in the atmosphere and significantly affects material examination in the THz range, was reduced [35]. During the measurements, the central parts of the samples were exposed to the broadband THz radiation, limited by the aperture having a diameter of 13 mm. For the purpose of this study, the refractive index and absorption coefficient were investigated as the crucial optical parameters for the design of DOEs. During the analysis stage, Blackman–Harris 4-term apodization filtration was applied to mitigate the effects of parasitic Fresnel reflections (at the interfaces between two different media), which are commonly observed in parallel samples [36]. Measurements were conducted on two samples for each setting of the investigated printing parameters, with most samples measured twice. After completing the measurements within each group, the mean value of the results for samples printed with the same parameters (typically four measurements) was calculated. Exemplary THz TDS measurement results of the reference samples, prior to the mean value calculation, are presented in Figure 4.

The analysis of the results in Figure 4 did not reveal significant differences between individual measurements; however, such differences are present. Errors and variations in THz TDS measurements stem from various random and systematic factors, including Fabry–Perot reflections within the samples, electronic and optical noise, inaccuracies in measuring sample thickness, the misalignment of the sample in the setup, fluctuations in the refractive index of air, jitter in the delay stage, and the impact of environmental factors such as temperature and humidity. Furthermore, the stability of the femtosecond laser and signal intensity fluctuations are crucial aspects. A detailed description of these factors can be found in the following studies: [37,38].

## 4. Experimental Results and Discussion

The measurement results were categorized based on the obtained outcomes. Figure 5 focuses on the characteristics where a significant impact on the THz optical properties with the change in printing parameters was not observed. Figure 6 presents the results where the influence on the THz optical parameters was observed. In Figure 5 and Figure 6, the solid lines correspond to the absorption coefficient α of the samples, and the dashed lines correspond to the refractive index *n* of the samples. The expected improvement in optical properties involves a reduction in the absorption coefficient while simultaneously increasing the refractive index. Refractive index values corresponds to the ones enabling reasonable thickness of the structures and low Fresnel losses.

The data analysis shown in Figure 5 indicates no significant influence of the printing temperature on the THz optical parameters of the samples (Figure 5a). A similar behavior was observed for the examination of three different sizes of nozzles (Figure 5b), where only an insignificant decrease in the refractive index for the 0.25 mm diameter nozzle was observed. Despite the higher printing precision using the smaller nozzle, the obtained result implies a decrease in the structural solidity of the printed object. The printing speed parameter in the range from 20 mm/s to 100 mm/s did not change the THz optical properties of the samples (Figure 5c). However, for the printing speed of 120 mm/s, a decrease in the refractive index with a slight increase in the absorption coefficient was observed. This indicates that for printing speeds that are too high, the precision of the print is reduced, and it results in the uneven distribution of individual lines composing the object. Then, in analyzing the influence of the various infill patterns on the THz optical behavior of the samples (Figure 5d), significant changes were not observed in the absorption coefficient of the samples. The sample manufactured in the vertical position as well as the samples that possess concentric and octagram spiral infill patterns exhibited lower refractive index values than the samples manufactured with rectilinear and aligned rectilinear patterns. It should be mentioned that the samples printed with rectilinear and aligned rectilinear patterns demonstrated very similar optical properties in the analyzed THz radiation range. While in visible light (the photographs shown in Figure 3), the sample with the aligned rectilinear pattern was significantly more transparent.

On the other hand, the characteristics of the THz optical properties of the examined samples manufactured with different 3D printing parameters are presented in Figure 6. Here, the samples where the change in the printing parameters introduced a significant impact on the optical properties were observed. Firstly, the choice of the proper extrusion width of the lines of the print is essential (Figure 6a). The line width should be larger than the diameter of the nozzle due to the non-Newtonian nature of polymers and the Barus effect occurring during polymer flow through the channel, which results in material bulging. The authors recommend a 0.45 mm extrusion width for a 0.40 mm nozzle diameter. Enforcing an excessively high width of extruded lines during the printing process (20% higher than the diameter of the nozzle) results in the production of deformed lines. Consequently, this leads to the increase in the absorption coefficient and a slight decrease in the refractive index of the printed structure.

Secondly, the parasitic impact of print cooling on the THz optical properties was noticed, as depicted in Figure 6b. With an escalation in the cooling level, a significant decrease in the refractive index for lower THz frequencies was observed. Additionally, the samples exhibited a significant increase in the absorption coefficient for higher THz frequencies. This phenomenon is attributed to a reduction in the homogeneity of the samples resulting from the incomplete bonding between layers caused by the cooling. When lower THz frequencies are taken into consideration, the layers of the structures with the air gaps between them are treated by the THz beam as single components, due to their sub-wavelength dimensions. Thus, the THz radiation recognizes the lower refractive index of the manufactured object. However, when the higher THz frequencies propagate through the sample, the THz beam interacts with the components (semi-separated/combined layers), which results in signal losses (e.g., due to the diffraction, interference, and dispersion effects as well as secondary Fresnel reflection losses). Therefore, the absorption coefficient of the manufactured object increases significantly. Consequently, the recommendation is to turn off the cooling for the printing of DOEs when it does not lead to deformations in the printed object.

A similar tendency but with enhanced effects was observed while analyzing the material flow ratio parameter, as presented in Figure 6c. For the lower frequencies when the material flow of the print is too low (90% or 95% of the standard value), the refractive index values are visibly smaller in comparison with other samples. Moreover, for the 2 THz frequency, the absorption coefficient was even three times higher compared to the reference structure. In this particular situation, the phenomenon refers to the gaps created between printed lines caused by the lack of material building the printed object. However, when the material flow ratio is too high (110% in Figure 6c), the worsening of the THz optical properties is also significant and is the result of the deformation of the sample illustrated in Figure 2e,f. This time, the increase in the absorption coefficient and the significant decrease in the refractive index were noticed. It should be emphasized that unlike the previously mentioned situation when the 110% material flow ratio was applied, in this case, the refractive index changes were independent of the frequency (similar to the whole radiation range analyzed) compared to the reference structure. The authors recommend a slightly increased material flow ratio parameter for the manufacturing of DOEs using COC material with the value ranging between 102% and 105% of the standard value. Therefore, the extended filling of the interline gaps is obtained, providing a more homogeneous printing and at the same time avoiding deformations of the printed elements.

A comparison of the THz optical behavior of the 3D printed samples with various layer thicknesses is presented in Figure 6d. It has been observed that the thickness equal to 100 μm or 150 μm is the optimal thickness for the line and consequently the layer thicknesses in 3D printing using FDM technology for THz DOEs. The decrease in the refractive index was also observed, with the increase in line thickness being relatively constant in the whole analyzed range. It needs to be mentioned that the absorption coefficient increased significantly for the line thicknesses above 200 μm. The results indicate higher homogeneity of the samples printed with smaller layer thicknesses. However, it should be highlighted that the samples in this analysis were manufactured without cooling the prints, and this factor might have a greater impact on the optical behavior of the samples containing a greater number of layers (the smaller line thicknesses). For the purpose of this study, only a single 3D printing parameter was changed in each series of the manufactured samples. This is due to the fact that if the 3D printing parameters deviate from the optimal settings, defects and delaminations occur in the manufactured samples. These defects are visible, as shown in Figure 2, and they adversely affect the optical properties of the printed objects, as demonstrated in this study. Moreover, printing such samples becomes challenging because successive lines and layers adhere poorly, leading to defects that often cause the print to fail. When multiple parameters are changed simultaneously in a way that negatively impacts the printing process, it becomes impossible to successfully complete the print. In such cases, the lines do not bond properly or may separate during printing, or the delaminations become so significant that the printing process cannot continue, resulting in failure.

## 5. Conclusions

In this study, the authors presented an investigation on the influence of various 3D printing parameters using COC material on the THz optical properties of manufactured structures in the THz radiation range. COC is a highly transparent biocompatible material that might be applied in systems where contact with living tissues is necessary. Polymer-based manufacturing processes play a crucial role not only in the development of THz optical systems but also in other innovative technological designs and optimization methods aiming to lower the environmental impact of polymer-based manufacturing processes. All of them help to preserve product quality and material performance in light of applications in medicine, agriculture, coatings and paints, biotechnology, automotive, packaging, textiles, electronics, and many more. Thus, enabling the creation of lightweight, durable, and versatile components such as versatile optical components is crucial, while ensuring product quality, specific optical properties, and cost-effectiveness. The FDM 3D printing technology used for the manufacturing of samples directly corresponds to the production/prototyping process of DOEs that might be implemented in the mentioned systems.

The results presented in this study indicate that some printing parameters have significant influence on the THz optical behavior of the manufactured structures. These parameters are the extrusion width of the lines, the printing cooling level, the material flow ratio, and the line (layer) thickness of the print. Inappropriately set manufacturing parameters result in the worsening of the THz optical properties of the structure, such as a significant decrease in the refractive index or a several-fold increase in the absorption coefficient. Furthermore, they may lead to the formation of defects in the printed structure. The authors recommend the following printing parameters for the manufacturing of DOEs using FDM technology with COC material: 400 μm for the nozzle diameter, 245–255 °C for the nozzle temperature, 60 mm/s for the printing speed (but 20 mm/s of the first layer of the print), 450 μm for the width of the extruded lines, no cooling (0% cooling level), 102–105% for the material flow ratio of the standard value, 100 μm for the layer height (150 μm also gives the desired results), and a rectilinear or aligned rectilinear infill pattern.

## Figures and Tables

**Figure 1 materials-17-05104-f001:**
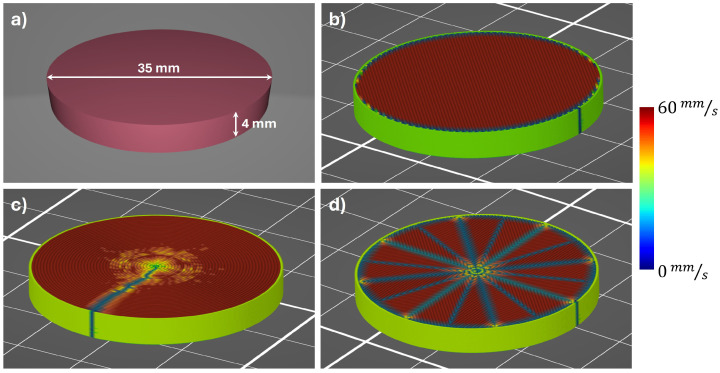
Spatial model of the cylindrical sample sliced using reference printing parameters with various infill patterns. The colors of the sliced objects indicate the predicted printing speeds achieved by the nozzle during the printing process. (**a**) Three-dimensional model of the sample; (**b**) sliced model with (aligned) rectilinear infill; (**c**) sliced model with concentric infill; (**d**) sliced model with octagram spiral infill.

**Figure 2 materials-17-05104-f002:**
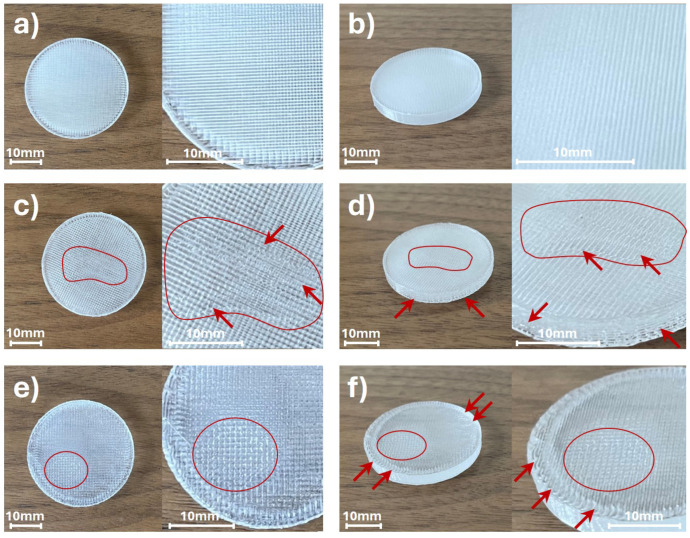
Exemplary 3D printed samples manufactured from COC material using FDM technology, subsequently examined with THz TDS; (**a**,**b**) reference samples without visible deformations; (**c**,**d**) samples manufactured with too-thick layers that result in deformations (marked with a red outline and red arrows) in the central part of the samples; (**e**,**f**) samples printed with an overly high material flow rate that results in overflowing material leading to the deformations (marked with a red outline and red arrows) visible on the top surface and the edges of the samples.

**Figure 3 materials-17-05104-f003:**
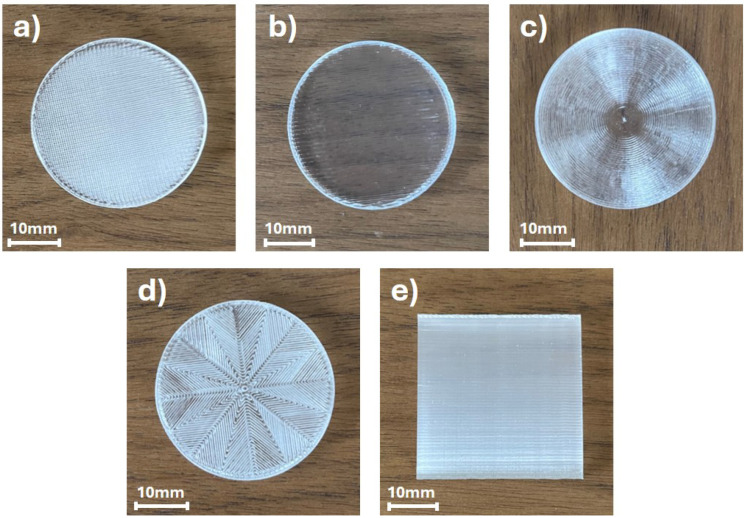
Three-dimensional printed samples manufactured from COC material using FDM technology, subsequently examined with THz TDS. The photographs correspond to the samples manufactured with different infill patterns: (**a**) rectilinear; (**b**) aligned rectilinear; (**c**) concentric; (**d**) octagram spiral; and (**e**) the cuboid sample printed in the vertical position with rectilinear infill pattern.

**Figure 4 materials-17-05104-f004:**
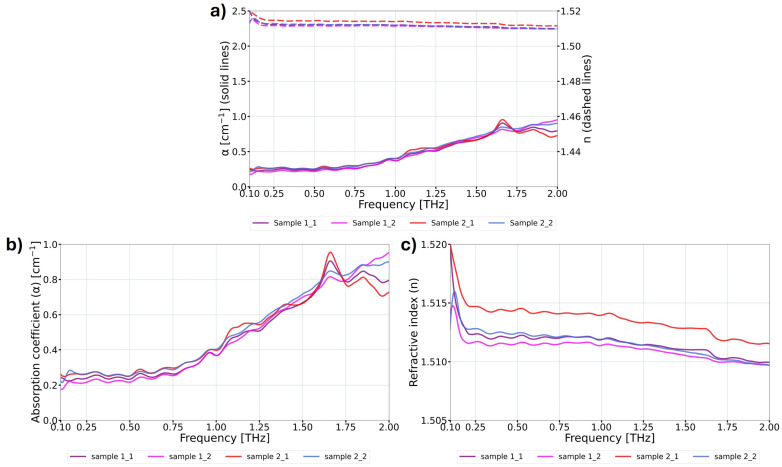
THz TDS measurement results obtained from two different reference samples, each measured twice at appropriate time intervals. Each measurement was performed with a corresponding reference measurement (signal without the sample) and covered different sample areas; (**a**) the absorption coefficient α (solid lines) and refractive index *n* (dashed lines) in the THz frequency domain, displayed over a broader data range common to all measurements presented in this study; (**b**) the absorption coefficient in the THz frequency domain; (**c**) the refractive index in the THz frequency domain.

**Figure 5 materials-17-05104-f005:**
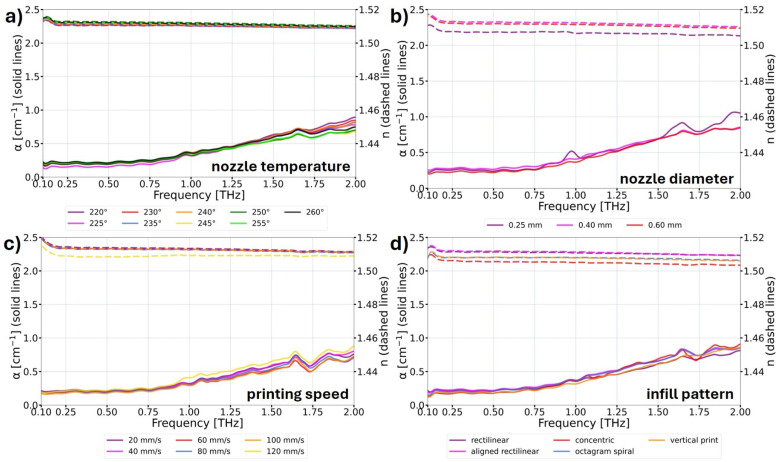
THz optical properties of COC samples 3D printed using FDM, illustrating the impact of changes in various printing parameters. The data show the measured absorption coefficient α (solid lines) and refractive index *n* (dashed lines) in the THz frequency domain, indicating the influence of the following printing parameters: (**a**) nozzle temperatures; (**b**) nozzle diameters; (**c**) printing speed; (**d**) infill patterns. The change in these parameters does not change the THz optical parameters of the samples.

**Figure 6 materials-17-05104-f006:**
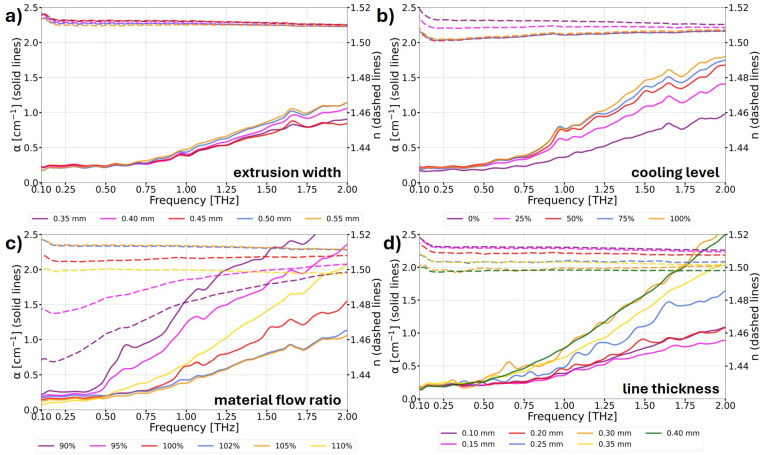
THz optical properties of COC samples 3D printed using FDM, illustrating the impact of changes in various printing parameters. The data show the measured absorption coefficient α (solid lines) and refractive index *n* (dashed lines) in the THz frequency domain, indicating the influence of the following printing parameters: (**a**) extrusion width; (**b**) cooling level; (**c**) material flow ratio; (**d**) line thickness. The change in these parameters introduce changes in the THz optical parameters of the samples.

## Data Availability

The raw data supporting the conclusions of this article will be made available by the authors on request.

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
