# Peer review of "Exploring the Impact of 3D Printing Parameters on the THz Optical Characteristics of COC Material"

_materials, 2024, doi:10.3390/ma17205104_

Round 1
Reviewer 1 Report
Comments and Suggestions for Authors
The authors show the 3d printing of a cyclic olefin copolymer (COC) for the application in THz optics. They tested different printing parameters and their influence on the optical parameters of the printed strands. The introduction and requirements are well described.
The materials are well described, and the study is well designed. I just miss some crucial information. I found some of them in the two sources 16 and 19 mentioned by you, but even there, there was e.g. no mentioning of the printer used. So I suggest adding the following information to the manuscript or adding references pointing to exactly appropriate sources to find the information:
- Which printer in which configuration was used?
- Which slicer in which version was used?
- What is the printing material (filament) manufacturer and article number or name?
- How many samples per printing parameter were printed and investigated with THz TDS?
- Which instrument and settings were used for the THz TDS?
The results and discussion are well written and sound. But I have some questions:
- How is the variation of the optical properties within samples printed with the same printing parameters? So, how do the measured optical properties vary, when printing and measuring the same sample several times?
- Are the THz optical properties shown in figures 3 and 4 per printing parameter from one sample measured once, or do the plots show a mean value over several samples? In the latter: What is the error, e.g. standard deviation?
Comments on the Quality of English LanguageI recommend a check of typos as there are some small mistakes such as e.g.:
- Line 61: “…diffractive optics is increasingly being utilized…” I assume this should be an “are” instead of “is”
- Line 86: “…to design relatively thin element without...” I assume this should be “elements”
Reviewer 2 Report
Comments and Suggestions for Authors
This paper presents an experimental investigation on some process parameters that may affect the optical properties within the THz radiation range of cyclic olefin copolymer (COC) structures manufactured by FDM. The analysed factors are the nozzle temperature, the nozzle size, the printing speed, the extruded line width, the cooling fan speed, the material flow ratio, the layer thickness, and the infill pattern type. The topic of this study is of potential interest for the additive manufacturing and optics communities, but the paper has to be significantly improved to be considered for publication in a journal.
The reported state of the art is not effective in highlighting the new knowledge introduced by this study compared to the existing literature. The papers cited at lines 71, 79 and 84 have to be described in more detail and their findings have to be commented, otherwise it is not clear if the study novelty consists of:
§ the manufacturing process used to manufacture THz optical elements,
§ the used material,
§ the performed analysis,
§ … … …
The technical treatment is poor because many details of the experimental procedures and experimental design are missing.
In particular, you have to provide more details about the FDM printer (e.g. nozzle diameter, XY accuracy, Z resolution, etc.), the COC filament (manufacturer, code, diameter), the slicer software.
Based on Section 3, the experiments were performed following a “one-factor-at-a-time” approach. You should explain why you selected this approach, which is disadvantageous with respect to a factorial experimental design (e.g. a 2k factorial design for factor screening), as it does not allow investigating the possible effect of an interaction between the selected factors.
DETAILED REVIEW
3. Materials
- pag 3, lines 134-135: Since this paper is not particularly long, you might consider adding the 3D models of the structures.
- pag 4, fig 1:
o Add the scale bar in each picture.
o In the pictures of fig. 1b, 1d and 1f the samples are not really seen from the side.
o It is not easy to see the defects described in the text, you should add boxes/circles/arrows to highlight them.
- pag 5, fig 2: Add the scale bar in each picture.
4. Experimental Results and Discussion
- pag 5, lines 187-197: Section 3 is more appropriate for this content.
- pag 6, fig 3 and pag 7, fig 4: Since this paper is not particularly long, you might consider showing separate graphs for the absorption coefficient a and refractive index n.
Round 2
Reviewer 2 Report
Comments and Suggestions for Authors
The paper has been improved compared to the first submitted version, but it could be further enhanced.
[plain text = my previous comments, blue = my new comments]
The reported state of the art is not effective in highlighting the new knowledge introduced by this study compared to the existing literature. The papers cited at lines 71, 79 and 84 have to be described in more detail and their findings have to be commented, otherwise it is not clear if the study novelty consists of:
§ the manufacturing process used to manufacture THz optical elements,
§ the used material,
§ the performed analysis,
§ … … …
Section 2 can be further improved by explicitly stating which AM processes are used in the papers in lines 71-75 and for the materials listed in lines 82-84, especially for COC.
The technical treatment is poor because many details of the experimental procedures and experimental design are missing.
In particular, you have to provide more details about the FDM printer (e.g. nozzle diameter, XY accuracy, Z resolution, etc.), the COC filament (manufacturer, code, diameter), the slicer software.
Based on Section 3, the experiments were performed following a “one-factor-at-a-time” approach. You should explain why you selected this approach, which is disadvantageous with respect to a factorial experimental design (e.g. a 2k factorial design for factor screening), as it does not allow investigating the possible effect of an interaction between the selected factors.
In your reply you state that “no significant interactions were observed between the parameters, except for layer thickness and cooling level”. How could you study the interactions if you chose to “investigate the impact of each parameter individually” ?
